# Transporter-Mediated Mitochondrial GSH Depletion Leading to Mitochondrial Dysfunction and Rescue with αB Crystallin Peptide in RPE Cells

**DOI:** 10.3390/antiox9050411

**Published:** 2020-05-12

**Authors:** Parameswaran G Sreekumar, Mo Wang, Christine Spee, Srinivas R. Sadda, Ram Kannan

**Affiliations:** 1The Stephen J. Ryan Initiative for Macular Research (RIMR), Doheny Eye Institute, DVRC 203, 1355 San Pablo Street, Los Angeles, CA 90033, USA; sparameswaran@doheny.org (P.G.S.); mwang@doheny.org (M.W.); cspee@doheny.org (C.S.); ssadda@doheny.org (S.R.S.); 2Department of Ophthalmology, David Geffen School of Medicine, University of California-Los Angeles, Los Angeles, CA 90095, USA

**Keywords:** mitochondrial membrane anion transporters, GSH carriers, αB crystallin peptide, bioenergetics, polarized RPE

## Abstract

Mitochondrial glutathione (mGSH) is critical for cell survival. We recently reported the localization of OGC (SLC25A11) and DIC (SLC25A10) in hRPE. Herein, we investigated the suppression of OGC and DIC and the effect of αB crystallin chaperone peptide co-treatment on RPE cell death and mitochondrial function. Non-polarized and polarized human RPE were co-treated for 24 h with phenyl succinic acid (PS, 5 mM) or butyl malonic acid (BM, 5 mM) with or without αB cry peptide (75 µg/mL). mGSH levels, mitochondrial bioenergetics, and ETC proteins were analyzed. The effect of mGSH depletion on cell death and barrier function was determined in polarized RPE co-treated with PS, OGC siRNA or BM and αB cry peptide. Inhibition of OGC and DIC resulted in a significant decrease in mGSH and increased apoptosis. mGSH depletion significantly decreased mitochondrial respiration, ATP production, and altered ETC protein expression. αB cry peptide restored mGSH, attenuated apoptosis, upregulated ETC proteins, and improved mitochondrial bioenergetics and biogenesis. mGSH transporters exhibited differential polarized localization: DIC (apical) and OGC (apical and basal). Inhibition of mGSH transport compromised barrier function which was partially restored by αB cry peptide. Our findings suggest mGSH augmentation by its transporters may be a valuable approach in AMD therapy.

## 1. Introduction

Age-related macular degeneration (AMD), the leading cause of irreversible visual impairment, is a process by which the structure and function of the macula gradually deteriorate, and symptoms become clinically evident in people 50 years or older [1]. While AMD is a complex and multifactorial disease, the dysfunction of retinal pigment epithelium (RPE) is considered to play a key role [2,3,4]. RPE forms a quiescent monolayer of non-proliferating cells, strategically located between the choriocapillaris/Bruch’s membrane complex and the light-sensitive photoreceptors. Oxidative damage has been implicated in AMD pathogenesis [5]. The clinical hallmark of AMD is the appearance of hard or soft drusen [6], which are localized yellowish deposits of oxidized lipids, proteins, and inflammatory debris lying between RPE and Bruch’s membrane. Drusen, the predominant clinical characteristic of early AMD, is derived from RPE and interrupts the RPE monolayer. Drusen constituents are implicated in AMD pathogenesis such as complement factors, amyloid β, and double-stranded RNA, inducing inflammatory and degenerative effects in RPE [6]. Much of the RPE toxicity has been attributed to reactive oxygen species (ROS) formation [2,7]. RPE dysfunction leads to cell death of photoreceptors and consequent irreversible vision loss.

One of the primary contributing factors for macular degeneration is oxidative stress, which refers to cellular damage caused by ROS, a process that has also been implicated in many degenerative diseases. Mitochondria generate ROS as a byproduct of oxidative phosphorylation, and increased mitochondrial ROS damages mitochondria and, along with the action of cytosolic ROS, leads to mitochondrial dysfunction [4]. The mitochondrial damage and dysfunction disrupt the bioenergetic metabolic pathway by reducing mitochondrial energy production due to reduced oxygen consumption [4,8]. In this context, it is becoming increasingly evident that the redox status of RPE cells plays a critical role in combating oxidant stress. 

GSH is one of the most predominant antioxidant molecules in RPE cells and is present at high concentrations in the retina and RPE [9,10]. GSH protects against oxidative damage in many tissues, including RPE [9,11]. Exogenously administered GSH or GSH ester protects against oxidative damage in cultured human RPE, while GSH depletion was shown to cause cell death [12]. Previous work from our laboratory has shown that mitochondrial GSH (mGSH) plays a critical role in RPE cell survival [9,11,13]. GSH transport mechanisms of different cellular compartments of tissues including the retina have received considerable attention in recent years. Our laboratory demonstrated the role of multidrug resistance-associated protein 1 (MRP1) in GSH and GSSG efflux in RPE cells [9]. However, unlike cytosol, mitochondria do not contain the enzymatic machinery to synthesize GSH from its constituent amino acids [14,15]. Previous studies in lung and heart tissues provided evidence that two inner mitochondrial membrane anion transporters, the dicarboxylate carrier (DIC, SLC25A10), and the 2-oxoglutarate carrier (OGC, SLC25A11) transport GSH into mitochondria [15,16,17]. However, very little is known on the expression and regulation of mGSH transporters in RPE cells and the retina. Very recently, our laboratory characterized and localized OGC and DIC transporters to mitochondria in primary RPE cells [11]. Selective inhibition of OGC and DIC resulted in marked mGSH depletion and caused significant RPE cell death [11]. This study addresses the effect of regulation of the mGSH transporters on mitochondrial function in primary RPE cells and in polarized RPE monolayers and explores modalities to prevent mitochondrial dysfunction by treatment with a short-chain length 20-mer peptide of αB Crystallin. 

Among the small heat shock proteins, αB crystallin is highly expressed in the RPE cells [18,19,20]. αB crystallin acts as a molecular chaperone, inhibiting oxidative stress-induced cell death, preventing aggregation of proteins and inflammation [19]. Many short-chain peptides from αB crystallin have been identified [21,22]. Among them, a 20-mer functional chaperone peptide (αB cry peptide) derived from the amino acid residues 73–92 (DRFSVNLDVKHFSPEELKVK) of αB crystallin protects RPE cells and lens epithelial cells from oxidative stress-induced cell death by inhibiting caspase-3 activation both in in vitro [20,23,24,25] and in in vivo models [25,26,27]. In addition, overexpression of αB crystallin significantly increased cellular GSH in RPE and a prominent increase in mGSH [9]. In the present study, we have investigated the mechanisms by which mGSH helps maintain RPE cell viability with special emphasis on the mechanism relating to mitochondrial bioenergetics, biogenesis, and barrier function. Furthermore, we have investigated the antiapoptotic rescuing function of αB cry peptide under conditions of mGSH deficiency in RPE cells. 

## 2. Materials and Methods 

### 2.1. Cell Culture and Treatment 

All experiments were conducted in compliance with the tenets of the Declaration of Helsinki and ARVO guidelines. Human retinal pigment epithelium (RPE) cells were isolated from human fetal eyes (gestational age 16–18 weeks) obtained from Novogenix Lab (Los Angeles, CA, USA). Primary cultures of hRPE cells and polarized RPE monolayers were established as described previously [28,29]. Second to fourth passage RPE cells were used in all experiments. In brief, the hRPE cells were grown in Dulbecco’s modified Eagle medium (DMEM, Fisher Scientific, Pittsburgh, PA, USA) with 10% fetal bovine serum (FBS, Laguna Scientific, Laguna Niguel CA, USA), and the cells were confluent at the time of treatment with specified inhibitors described below. The protocol for the generation of long-term polarized RPE cultures was described in our earlier publications [29,30]. 

To study the blockade of DIC and OGC expression by chemical inhibitors, confluent RPE cells were incubated with 5 mM phenylsuccinic acid (PS) and 5 mM butylmalonic acid (BM; Sigma-Aldrich Corp., St. Louis, MO, USA) for 24 h. To assess the effect of αB cry peptide, cells were co-treated with 5 mM PS or BM, in the presence 75 µg/mL αB cry peptide (DRFSVNLDVKHFSPEELKVK; Neo-peptide, Cambridge, MA, USA) in serum free medium. We also studied the effect of blocking OGC and DIC in polarized RPE monolayers. We had previously shown that polarized RPE cells are highly resistant to stress [27,30,31] and hence in the present study we used higher doses of PS (30 mM) or BM (30 mM). 

### 2.2. Protection of RPE by Exogenous αB Cry Peptide from PS- or BM-Induced Cell Death

The effect of co-treatment with αB cry peptide was studied in confluent human RPE cells challenged with either BM or PS alone or together with 75 µg/mL αB cry peptide for 24 h. Cell death was detected by the TUNEL assay following the manufacturer’s protocol (In Situ Cell Death Detection Kit; Roche, IN, USA). In short, RPE cells were treated with inhibitors (BM or PS), in the presence or absence of αB cry peptide for 24 h. The number of TUNEL positive cells were counted under a Keyence fluorescence digital microscope (Keyence, Itasca, IL, USA) and presented as the percentage of dead cells [31].

### 2.3. siRNA-Mediated Knockdown of OGC

hRPE cells at 50–60% confluence was used for transfection studies. The siRNA targeting human OGC sequences (HSS112214) (Invitrogen, Carlsbad, CA) and control siRNA (Santa Cruz Biotechnology, Inc., Dallas, TX USA) were mixed with RNAi MAX transfection reagent (Life Technologies, Carlsbad, CA)) [11]. To avoid cytotoxicity, the transfection medium was replaced with a complete medium at 6 h after siRNA transfection. OGC mRNA and protein expression were analyzed by real-time RT-PCR and immunoblot analysis after 24 h and 48 h post-transfection, respectively. To study the effect of OGC knockdown and effect of αB cry peptide on mitochondrial respiration or cell death, the OGC silenced cells were incubated with 75 µg/mL αB cry peptide 24 h and assayed for cell death and respiration. 

### 2.4. Caspase 3/7 Activation Using IncuCyte Cell Apoptosis Assay

hRPE cells (5500 cells/well) plated in 96-well plates were used. Confluent cells were co-treated with PS (5 mM) or BM (5 mM) or PS +75 µg/mL αB cry peptide or BM + 75 µg/mL αB cry peptide. Caspase 3/7 reagent SYTOX Green diluted in cell culture media (1:1000 dilution) to make a total volume of 100 μL/well. Each treatment condition was performed in 5–8 wells. Cell apoptosis was monitored for 24 h using a live cell analysis system (IncuCyte ZOOM; Essen Bioscience, Ann Arbor, MI, USA) as described earlier [11]. 

### 2.5. Isolation of Mitochondrial and Cytosolic Fractions and Assay for GSH 

Isolation of mitochondrial and cytosolic proteins from RPE cells was carried out following the previously described procedure [11,13]. Protein concertation was determined using a commercial kit (Bio-Rad, Hercules, CA, USA). An equal amount of protein was loaded into 96-well plates, and the total cellular, mitochondrial, and cytosolic GSH levels were determined with a colorimetric glutathione assay kit following the manufacturer’s protocol (BioVision, Milpitas, CA). Cellular GSH concentrations were expressed as microgram/10^6^ cells and were normalized to percent of control. Data presented were based on at least three independent experiments, and each experiment was performed in triplicate.

### 2.6. Measurement of Cellular Respiration

Mitochondrial bioenergetics was determined by measuring the oxygen consumption rate (OCR) of RPE cells with a Seahorse XFe96 Analyzer (Agilent, Santa Clara, CA, USA) as described previously [8]. Confluent RPE cells were treated for 24 h with either PS or BM alone or BM+ mini cry or PM+ αB cry peptide (75 μg/mL). Assays were initiated by replacing the growth medium with 175 μL XF assay medium. XF assay medium contained glucose (25 mM), sodium pyruvate (1 mM), and glutamine (2 mM) (Agilent). The concentration of inhibitors was oligomycin (ATP-Synthase inhibitor) at 1.5 µM; carbonyl cyanide 4-(trifluoromethoxy) phenylhydrazone (FCCP, mitochondrial membrane depolarizer) at 1.5 µM and, a mixture of 0.5 µM of each rotenone (complex I inhibitor) and antimycin A (complex III inhibitor). We calculated OCR-linked ATP production, maximal respiration capacity, and spare respiratory capacity, and basal respiration. Each sample was measured in five to ten wells per condition, and the results were averaged. The experiments were repeated three times and the mean values averaged to achieve *n* = 3 to 4 per condition. RPE cells were transfected with OGC siRNA and after 48 h cells were seeded at 30,000 cells per well in Seahorse XF96 tissue culture plates. After 24 h, cells were incubated with 25 or 75 µg/mL αB cry peptide and incubated for 24 h. The OCR data were expressed as pmol/min/μg protein.

### 2.7. Western Blot Analysis

Protein was extracted from cells with RIPA buffer containing protease inhibitor and concentration of soluble protein was measured using BSA as standard. Equal amounts of protein (30 µg) were resolved on TGX-precast gels (Bio-Rad, Hercules, CA, USA) and transferred to PVDF blotting membranes (Millipore, Billerica, MA, USA). Membranes were probed with respective primary antibodies overnight at 4 °C (see Table 1 for a list of antibodies). After incubation with the appropriate secondary antibodies (Vector Laboratories, Burlingame, CA, USA), protein bands were visualized by a chemiluminescence (ECL) detection system (Thermo Fisher Scientific, IL, USA). Equal protein loading was confirmed with β-actin.

### 2.8. Localization of DIC and OGC in Polarized RPE Monolayers

RPE monolayers grown on Transwell filters [31]) were fixed in 4% PFA, and subsequently permeabilized with 0.1% Triton-X 100 for 15 min. Samples were blocked in 5% normal goat serum followed by incubating with either OGC (1:100) and DIC (1:100) rabbit polyclonal antibodies overnight at 4 °C. The cells were washed and incubated with fluorescein conjugated secondary antibody (Vector Labs, Burlingame, CA, USA) for 30 min at room temperature. Transwell membranes were cut and removed from the inserts with a fine razor and mounted on a microslide. The specimen was viewed on an LSM 770 laser-scanning microscope and (Carl Zeiss, Thornwood, NY, USA).

### 2.9. Polarity of the Expression of DIC and OGC in Polarized RPE Monolayers 

Polarized RPE monolayer was fixed in 4% PFA followed by permeabilization and blocking [31]. Cells were incubated with DIC (1:100) or OGC (1:100) antibodies overnight at 4 °C. After the immunostaining procedure, membranes were removed from the inserts with a sharp, razor by inserting it at one side of the filter and then gently moving it around the filter. Sample images were obtained on a confocal microscope (LSM 770) with a ×40 objective. Serial (0.5 μm) x–y (en face) x–z (top to bottom) sections were collected and processed. Images presented are representatives of confocal x–y and x–z sections.

### 2.10. Transepithelial Resistance in Polarized RPE and Expression of Tight Junction Protein

We have previously described the morphologic features and evidence for the integrity of polarized RPE monolayers that included transepithelial resistance (TER) and tight junction proteins ZO1, occludin, and Na/K ATPase [29,31]. In the present study, we examined the effect of inhibition of GSH transporters on highly polarized RPE cells on TER and tight junction protein ZO1. Polarized RPE monolayers were treated with either 30 mM PS or 30 mM BM with or without cotreatment with 200 µg/mL αB cry peptide for 24 h. TER was measured using an epithelial volt-ohm meter (EVOM; World Precision Instruments Inc., Sarasota, FL, USA) before and after the treatment. The cells were fixed in 4% PFA, blocked with 5% goat serum followed by incubation with ZO-1 rabbit polyclonal antibody overnight. After washing, cells were incubated with fluorescein-conjugated anti-rabbit secondary antibody (Vector Laboratories) for 30 min. After subsequent washing, membranes were cut with a scalpel, mounted on micro slides and samples were viewed under confocal microscopy (LSM 710, Carl Zeiss, Thornwood, NY, USA).

### 2.11. Statistical Analysis

All data are expressed as mean ± SEM. Data were analyzed using one-way ANOVA followed by Tukey post-test using graphing software (GraphPad Prism, version 5; GraphPad Software, Inc., La Jolla, CA, USA). *p* < 0.05 was considered significant.

## 3. Results

### 3.1. Inhibition of Mitochondrial GSH Carrier Protein Triggers RPE Apoptosis

Mitochondrial GSH is critical for regulating the redox status of the cells and, since mitochondria lack GSH synthetic machinery, the role of carrier proteins is crucial. Transport of GSH from the cytosol into the mitochondrial matrix is believed to be the sole mechanism that sustains the mGSH. It has been confirmed that, out of the eleven protein carriers that are known to reside in the inner mitochondrial membrane, the DIC and OGC act as main mitochondrial GSH transporters [16,32,33]. To study the role of OGC and DIC-mediated GSH transport and their role in cell protection, we examined cell death using TUNEL assay after blocking the transporters. We found that, in comparison with untreated cells, cells treated with inhibitors had significantly higher levels of cell death (*p* < 0.001 vs. control). We had previously shown that a 20-mer (αB cry peptide) from the C-terminal of αB crystallin has antiapoptotic properties [23]. In our experiments to test whether this peptide restores cell viability, we co-treated cells with 75 µg/mL αB cry peptide and DIC or OGC inhibitors. As expected, cell death was significantly reduced in the peptide-treated cells when compared to inhibitor only treated cells which confirmed the antiapoptotic function of αB cry peptide (Figure 1A,B).

To further confirm the findings obtained with pharmacological inhibitors, inhibition experiments after OGC silencing were performed. Cells in which OGC were silenced >75% (vs. control) were used for this purpose. OGC silencing rendered RPE cells susceptible to cell death as observed with chemical inhibitors (Figure 2A,B) and treatment with αB cry peptide significantly (*p* < 0.01) inhibited cell death (Figure 2A,B). 

### 3.2. Inhibition of mGSH Transport Activates Caspase 3/7

Since we found increased cell death after blocking mGSH carriers, real-time analysis of caspase 3/7 activation upon mGSH inhibition and effect of αB cry peptide was studied at two-time points, namely 6 and 13 h. The live-cell analysis system (Essen Bioscience) allows for real-time monitoring of cell apoptosis by determining the number of caspase 3/7 positive cells (green fluorescent labeling) at various treatment intervals. As shown in Figure 3, caspase 3/7 activity increased significantly at 6 and 13 h with either OGC or DIC inhibition. Co-treatment with 75 µg/mL αB cry peptide inhibited caspase activation which was statistically significant (*p* < 0.01 vs. PS or BM treated cells). No significant change in caspase 3/7 was observed in control cells at all the time points (Figure 3).

### 3.3. Inhibition of OGC or DIC Decreases Selectively the Mitochondrial GSH

Given the inhibition of OGC and DIC in RPE cells resulted in apoptosis and caspase 3 activation, we next addressed how their inhibition affects mitochondrial and cytosolic GSH pools. Treatment of RPE cells with BM or PS targeting OGC and DIC, respectively resulted in significant (*p* < 0.01) mGSH depletion compared to control cells (Figure 4A), suggesting that RPE cells rely on OGC and DIC to maintain mGSH. However, BM or PS treatment did not significantly affect cytosolic GSH levels (Figure 4B). Cotreatment with αB cry peptide replenished the levels of mGSH to that of control cells under conditions of OGC or DIC inhibition (Figure 4B), suggesting the increased cell survival observed with αB cry peptide treatment may be partly due to increased mGSH.

### 3.4. Pharmacological Inhibition of mGSH Transporters Alters Mitochondrial Bioenergetics 

Mitochondria are the sites of enzymatic reactions fundamental for life, producing most of the cellular ATP in eukaryotic cells by oxidative phosphorylation [34]. Since inhibition of DIC and OGC reduced mGSH pool by ∼60%, we conducted studies on the effects of mGSH depletion on mitochondrial respiratory parameters. Furthermore, studying detailed profiling of mitochondrial bioenergetics provides a comprehensive view of the nature of antioxidative and protective properties of αB cry peptide. As shown in Figure 5A,B, both OGC and DIC inhibition significantly decreased basal respiration following either OGC or DIC inhibition compared to control cells (Figure 5C). ATP production and proton leak were also significantly decreased (Figure 5E). A decrease in ATP production would indicate either low ATP demand or severe damage to the mitochondrial electron transport chain (ETC) [35]. These findings support the hypothesis that a decrease in mGSH pool downregulates mitochondrial oxygen consumption and ATP production and exacerbates cell death. Co-treatment with αB cry peptide significantly (*p* < 0.01 vs. BM or PS treated cells) increased basal respiration, maximal respiration, and ATP production. A significant decline in proton leak and rebound to control values with αB cry peptide was also observed. 

We also performed the bioenergetics experiments after silencing OGC in transfecting RPE cells with siRNA specifically targeting OGC (Figure 6). The result obtained was similar to those obtained with pharmacological inhibition of OGC with significantly decreased basal respiration, ATP production, and maximal respiration and αB cry treatment increased all those parameters (Figure 6 A–E).

### 3.5. Inhibition of GSH Transport Impairs ETC Protein Expression

Based on our bioenergetics data (Figure 5 and Figure 6), it seemed likely that cells treated with PS or BM have an impaired mitochondrial metabolism. To explore this in detail, we examined the expression of complex proteins (I–V), components of the electron transport chain. As anticipated, a decrease in the protein expression of complex proteins I and II (Figure 7A) was found with mGSH depletion and this decrease in expression was restored in αB cry peptide treated cells. However, the changes found in complex I protein expression have to be interpreted with caution because of the closely eluting bands for I and IV (Figure 7A). Our findings indicate that a reduction in the expression of mitochondria complex proteins is linked to decreased mitochondrial bioenergetics which can be attributed to mGSH deficiency following a blockade of mGSH transporters.

Since there was an increase in mitochondrial bioenergetics with αB cry peptide treatment, we investigated whether this is a result of increased mitochondrial biogenesis. Therefore, we evaluated the protein expression of mtTFA by Western blot analysis. An upregulation of mtTFA in cells co-treated with PS or BM and αB cry peptide was observed (Figure 7B), suggesting increased biogenesis with αB cry peptide. We then determined mitochondrial morphology in cells treated with PS or BM in the presence or absence of αB cry peptide. Staining with MitoTracker showed that inhibition of OGC or DIC resulted in a mitochondrial fragmentation phenotype (Figure 7C), which leads to an impairment in mitochondrial function. Co-treatment with αB cry peptide inhibited this phenomenon.

### 3.6. Polarized Localization of OGC and DIC in RPE Monolayers

The next set of experiments were designed to examine whether OGC and DIC transporters exhibit domain specificity in their expression. To examine the polarized cellular expression of the mitochondrial GSH transporters, we used primary RPE monolayers grown on Transwells. Immunofluorescence staining revealed the polarized expression of OGC and DIC in RPE monolayers in X–Y and X–Z selections (Figure 8). As shown in Figure 8, DIC is predominantly localized to the apical domain of the RPE, whereas OGC is distributed both at the apical and basolateral domains. 

### 3.7. Restoration of Barrier Properties of RPE Monolayers with αB Cry Peptide Treatment

We have previously demonstrated that polarized RPE monolayers are highly resistant to cell death due to increased secretion of growth factors and antioxidants [31]. In order to study the interrelationship between RPE junction integrity and mitochondrial GSH deficiency, we treated polarized monolayers (TER 471 ± 45 Ω.cm2) with PS (30 mM) or BM (30 mM) for 24 h. Treatment with PS or BM resulted in a significant (*p* < 0.001 vs. untreated controls) loss of TER (Figure 9A) caused by severe disruption in the pattern and loss of the tight junction protein, ZO1 (Figure 9B) and significant cell death (*p* < 0.001 vs. controls) (Figure 9C,D). Co-treatment with αB cry peptide resulted in the restoration of TER and protected RPE from apoptosis suggesting its antiapoptotic, therapeutic potential.

## 4. Discussion

An increasing number of pathological conditions are associated with marked depletion and or oxidation of mGSH, suggesting the importance of this pool [36,37]. mGSH accounts for about 10–15% of the total cellular glutathione pool. GSH enters the mitochondrial matrix via two anion transporters DIC and OGC localized on the inner mitochondrial membrane of cells. In the present study, we have shown that by modulating mitochondrial GSH transport through specific inhibition of DIC or OGC, RPE cells become more vulnerable to apoptosis. This can be attributed partly to mGSH depletion and treatment with αB cry peptide replenishes mGSH and improves cell survival. Furthermore, depletion of mGSH by pharmacological as well as by OGC siRNA significantly affects mitochondrial respiration and ETC proteins. In polarized RPE monolayers, mGSH depletion resulted in a significant decline in TER due to breaks in tight junction protein which could be prevented by αB cry peptide treatment. 

While RPE expresses several antioxidative proteins and enzymes [38], it is well known that GSH is the most prominent antioxidant in RPE cells and is present at a high concentration in the retina and RPE [13,31,39,40]. Severe effects of GSH depletion, mitochondrial dysfunction, and oxidative stress have been implicated in the pathology of a large number of neurodegenerative disorders, such as Alzheimer’s disease, Parkinson’s disease, amyotrophic lateral sclerosis, Huntington’s disease, and Friedreich’s ataxia, and AMD [4,41]. As mentioned earlier, mitochondria lack GSH synthetic machinery and since GSH cannot readily pass through lipid bilayer of the mitochondrial membrane, it relies on other mechanisms to enter mitochondria. The role of anion transporters OGC and DIC in mitochondrial GSH transport was first demonstrated by Lash and his coworkers [16] and the two transporters were characterized in RPE cells recently by our laboratory [11]. The importance of GSH in the suppression of mitochondrial ROS in stressed RPE cells was well established in our previous work [9,11,13]. Our present findings revealed that inhibition of mGSH transporters resulted in apoptosis in RPE cells and apoptotic cell death could partly arise from decreased mGSH levels. This is consistent with reports in multiple cell types that pharmacological inhibition or knocking down OGC or DIC resulted in significant downregulation of mitochondrial GSH [11,16,17,34,42]. In fact, studies have shown that cells can survive almost total loss of cytosolic GSH, but even a slight compromise in mGSH pool sensitizes them to MPT collapses and cell death [43,44]. 

Genetic studies using knockout mice of either OGC or DIC on the functional modification of tissues including the retina are lacking except for a few reports in the field of cancer. Recently, it was shown that knockout of OGC by the CRISPR/Cas 9 approach resulted in mitochondrial dysfunction from the inactivation of OGC leading to tumorigenesis [45]. Blocking OGC was found to reduce ATP production in another study and the authors concluded that OGC may have an advantage in arresting cancer growth [46]. As regards DIC, knockdown decreased NADPH production, regulated cell growth, and it was suggested that it could be a novel target in anti-cancer strategies [47]. The phenotype and morphology of retina under conditions of knockout of the two GSH carrier proteins remains to be determined.

While our focus in this work is on GSH, RPE cells also express redoxins and antioxidative enzymes such as Trx1, Trx2, Grx1, and Grx2 [38], and their contribution to cell survival cannot be overlooked. For example, it has been reported that the inactivation of the mitochondrial thioredoxin reductase (Trr2) causes early embryonic lethality, with increased liver apoptosis, reduced hematopoietic differentiation, heart defects with abnormal heart mitochondria, and slow-growing embryonic fibroblasts [48]. In Trr2-/-ic mice and in Trr2 deficient cardiac cells, GSH levels significantly decreased after oxidative stress, and this is most likely due to the inability of mitochondria to regenerate oxidized Txn2 in the absence of Trr2 [49]. At present, it remains unclear as to whether there is any synergistic interaction between mitochondrial anion transporters and Trr2. We observed significant caspase activation when mGSH transporters were inhibited and a significant reduction in mGSH levels. These findings agree with our previous observations that caspase 3/7 activation is rapid and occurs as early as 2 h [11] with significant change in mGSH. Furthermore, we report that co-treatment with 20-mer αB cry peptide reduced cell death by inhibiting caspase 3/7 activation and augmenting mGSH. The mechanistic aspects of interaction between αB cry peptide and anion transporters on mitochondrial function are unclear. In this context, it is worth mentioning that no specific receptor for αB crystallin or its peptide have been identified. Furthermore, our earlier functional studies have revealed that the αB cry peptide is transported into RPE cells by SOPT1 and SOPT2, two sodium-dependent oligopeptide transporters (SOPTs) [23]. We do not know at the present time whether or not αB cry peptide competes for mitochondrial anion transporter inhibitor binding site which will be interesting to study.

Previous work has established the chaperone function and antiapoptotic and anti-inflammatory properties of αB cry peptide in multiple cells and animal models [21,23,25,26,27,50]. It is to be noted that αB crystallin overexpression increases GSH content and confers resistance to oxidative stress in RPE cells, whereas silencing or knockout of αB crystallin led to GSH depletion, increasing oxidative stress both in vitro and in vivo [9,13]. Furthermore, the increase in GSH levels in αB crystallin overexpressing cells was found to be due to GSH biosynthesis as the gene expression of gamma-glutamyl cysteine synthetase, and the rate-limiting enzyme for GSH biosynthesis was also increased [9]. Since the percent GSH inhibition by chemical inhibitors of OGC and DIC is 60 to 70%, it is possible that there may be additional mechanisms that may be operative for GSH depletion in mitochondria. In this regard, there are some studies suggesting the role of Bcl-2 in the regulation of an essential pool of mitochondrial GSH; this regulation may depend on Bcl-2 directly interacting with GSH via the BH3 groove [51] or role of UCP2 in the transport of mGSH, but the mechanism is unclear [52]. It must be noted that this anion transporters also perform other functions. DIC exchanges dicarboxylates, such as malate and succinate, for phosphate, sulfate, and other small molecules, thereby providing substrates for metabolic processes including the Krebs cycle and fatty acid synthesis [53,54]. DIC is inhibited by Pi and other phosphate analogs [54] as well as substrate analogs such as alkyl malonates. Inhibition of DIC may also play a role in fatty-acid-mediated uncoupling [53]. OGC catalyzes the transport of 2-oxoglutarate across the inner mitochondrial membrane in an electroneutral exchange for the malate or other dicarboxylic acids and plays an important role in several metabolic processes [54]. OGC also may play a role in glucose-stimulated insulin secretion [55]. The OGC has also been suggested as the porphyrin transporter necessary for the mitochondrial import of the precursor porphyrin for final conversion to heme. Furthermore, OGC was shown to participate in the regulation of apoptosis and the regulation of mitochondrial morphology in *C. elegans* [56].

We further extended our studies to determine the importance of mGSH in regulating mitochondrial respiration in RPE cells. To the best of our knowledge, a mitochondrial anion inhibitor-induced change in mitochondrial respiration or protein import has not been reported. Our study with human RPE cells revealed that inhibition of OGC and DIC decreases OCR. Furthermore, the degree of restoration of OCR by co-treatment with αB cry peptide varied between OGC and DIC (Figure 6). The evidence that mGSH depletion negatively impacts mitochondrial respiration, ATP production, and reserve capacity suggests an oxidative stress-dependent mechanism, consistent with previous findings in other cell types showing the dependence of mitochondrial function and respiration on mGSH levels [57,58]. Both OGC and DIC inhibition caused a significant change in basal respiration, maximal O2 consumption, and respiratory reserve capacity, indicators of cellular bioenergetic resiliency. In line with previous reports, our results support a link between mitochondrial metabolism and redox homeostasis through mGSH status. Furthermore, our studies demonstrate the beneficial role of αB cry peptide on the respiratory parameters in RPE. Analysis of data on ETC protein expression suggested that ETC complex II was the main target site where both PS or BM exerted their inhibitory effect on the respiratory chain. Treatment of RPE cells with PS or BM caused a disruption of complex II. Our results on the expression pattern of complex I have to be interpreted with caution because of the close proximity of complex I and IV bands in WB analysis under non-reducing conditions. It was reported under acute redox cycling challenge to mitochondria that the complex I protein is more vulnerable [59]. However, a complex V segment of the respiratory chain was less impacted. Furthermore, while mGSH inhibition resulted in a significant inhibition of mitochondrial complex II expression, αB cry peptide treatment led to a marked restoration of expression to untreated levels suggesting that inhibition of complex II could contribute to dysregulated mito-bioenergetics in RPE cells. However, more detailed studies will be required to establish conclusively the role of ETC proteins in GSH dependent mitochondrial respiratory functions in RPE. 

The number of mitochondria within a cell is controlled by its turnover. To explore how decreased mGSH might modulate known components of the mitochondrial biogenesis machinery, we examined mTFA which is directly involved in the mtDNA replication in primary RPE cultures. In cells treated with PS or BM, mitochondrial biogenesis is greatly impaired, and this can be reversed by the addition of αB cry peptide. Direct evidence for impaired mitochondrial biogenesis in oxidative stress-induced RPE cells includes a significant decrease in mtDNA copy number and mTFA expression [8]. mTFA binds mitochondrial DNA and regulates mitochondrial transcription initiation, mtDNA copy number, packaging of mitochondrial DNA, and mitochondrial biogenesis [60,61]. Impaired mitochondrial biogenesis was also reported in RPE cells isolated from AMD donors as evidenced by decreased mtDNA copy number and ETC proteins [62,63]. 

Polarization is one of the salient features of the differentiated phenotype of the RPE monolayer and plays a key role in the vectorial transport of molecules. Furthermore, polarized RPE cells exhibit high TER and are mostly resistant to stress–induced cell death and only severe doses of oxidative stress can cause a break in tight junctions and a significant decrease in TER [8,31]. In the present study, we found polarized localization of DIC to the apical domain while OGC resides in both domains. The significance of these findings needs to be further explored particularly with respect to the influence of stress on vectorial distribution. As reported earlier, we used a higher concentration (>6-fold) of PS and BM to treat polarized cells. These doses resulted in tight junction protein break and a significant drop in TER and co-treatment with αB cry peptide restored both TER and the tight junction protein (Figure 9) This observation is in accordance with our recent findings in which co-treatment of stressed RPE with a mitochondrion derived peptide Humanin restored TER and improved cell viability of RPE [8].

In conclusion, we have found that mGSH is important for RPE survival and inhibition of the mGSH carrier proteins OGC and DIC results in mitochondrial dysfunction from impairment of mitochondrial biogenesis and bioenergetics. We further show that a chaperone peptide of αB crystallin can restore cell survival and may have potential as a valuable therapeutic agent in oxidative stress-induced diseases such as AMD. 

## Figures and Tables

**Figure 1 antioxidants-09-00411-f001:**
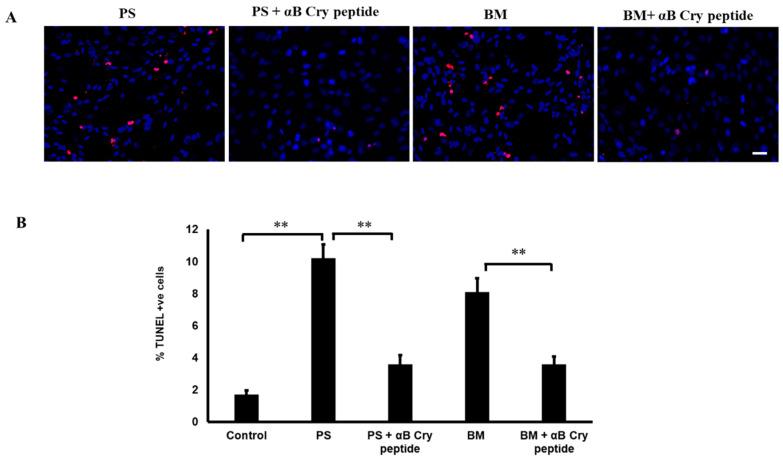
Increased RPE apoptosis with pharmacological inhibition of mGSH transporters OGC and DIC and attenuation of apoptosis by αB cry peptide treatment. (**A**) Primary cultured hRPE cells were treated with PS (5 mM) or BM (5 mM) with and without αB cry peptide (75 µg/mL) for 24 h. Apoptosis was determined by TUNEL staining (red). Nuclei were stained with DAPI (blue). (**B**). Quantification of TUNEL-positive cells is shown as % over controls. Co-treatment with αB cry peptide significantly attenuated RPE cell apoptosis caused by inhibition of mGSH transporters. Data are mean ± SEM. *n* = 3; ***p* < 0.01; Scale bar: 50 μm. mitochondrial glutathione (mGSH), human retinal pigment epithelium (RPE), Phenylsuccinic acid (PS), Butylmalonic acid (BM).

**Figure 2 antioxidants-09-00411-f002:**
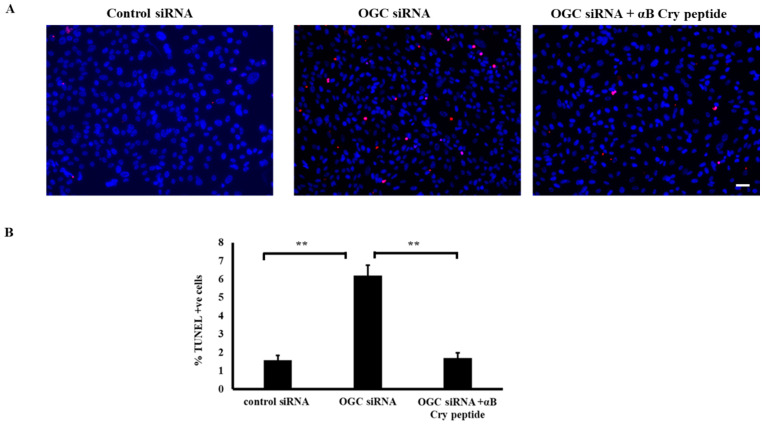
siRNA mediated knockdown of OGC augmented RPE cell death and αB cry peptide treatment increased cell survival. (**A**) Primary cultured RPE cells were transfected with OGC siRNA or control siRNA and 24 h post-transfected cells were treated with αB cry peptide (75 µg/mL) for an additional 24 h. Cell death was measured by TUNEL assay. The number of apoptotic cells increased 4-fold in OGC silenced RPE compared to control siRNA transfected group. (**B**) A significant reduction in apoptotic cells in αB cry peptide treated groups vs. inhibitor-treated groups was observed. *n* = 4; ***p* < 0.01; Scale bar: 50 μm.

**Figure 3 antioxidants-09-00411-f003:**
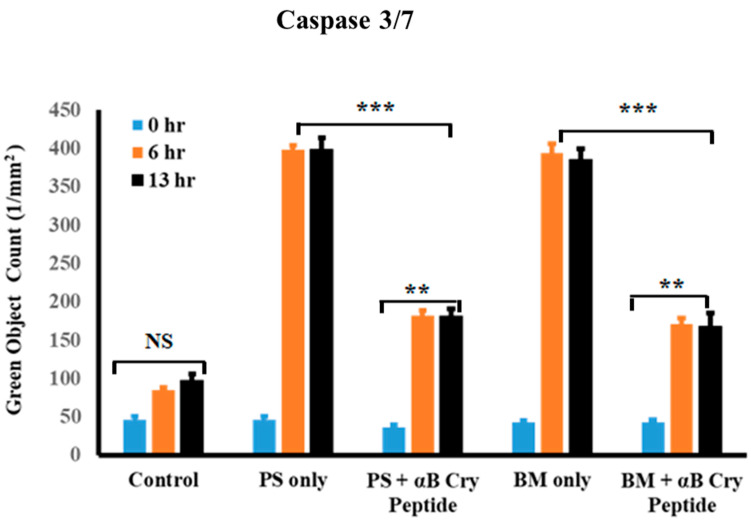
Inhibition of mGSH transporters in hRPE exacerbated caspase 3/7 activation. Real-time analysis of the number of caspase 3/7 positive cells was performed at 6 and 13 h post mGSH transporter inhibition with PS (5 mM) and BM (5 mM). Quantification of the activation of caspase 3/7 at 6 h and 13 h in the treatment groups showed significantly increased number of caspase positive cells. Coincubation with αB cry peptide (75 µg/mL) significantly inhibited caspase positive cells. Automated real-time assessment by live-cell analysis (Essen Bioscience), measured as green object count for all cells undergoing apoptosis have membrane compromise, and their DNA are stained with SYTOX Green dye. Quantification of the activation of caspase 3/7 (green fluorescence) at the indicated time points shows the suppression of inhibitor-induced caspase 3/7 activation by αB cry peptide. Data are shown as mean ± SEM. *n* = 4; ***p* < 0.01; ****p* < 0.001.

**Figure 4 antioxidants-09-00411-f004:**
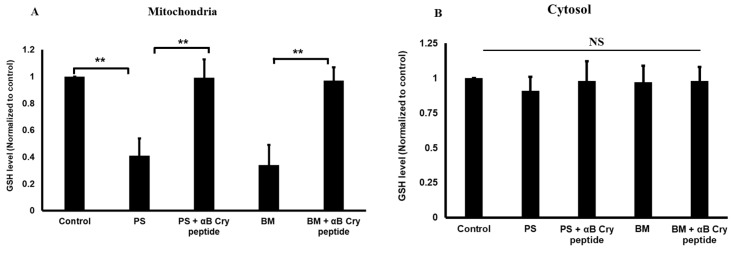
Inhibition of mGSH carrier proteins results in the depletion of mGSH levels of RPE. hRPE cells were incubated with OGC and DIC inhibitors in the presence or absence of αB cry peptide as described for Figure 1. Mitochondria and cytosolic fractions were separated, and GSH levels were measured in mitochondrial (**A**) and cytosolic fractions (**B**) by colorimetry and were normalized to controls. mGSH levels were significantly decreased with inhibitors only in the mitochondrial fraction while no significant changes were observed in the cytosol of RPE cells. Cotreatment with αB cry peptide restored mGSH levels to that of control levels. Data are shown as mean ± SEM of four independent experiments. ***p* < 0.01; NS = not significant.

**Figure 5 antioxidants-09-00411-f005:**
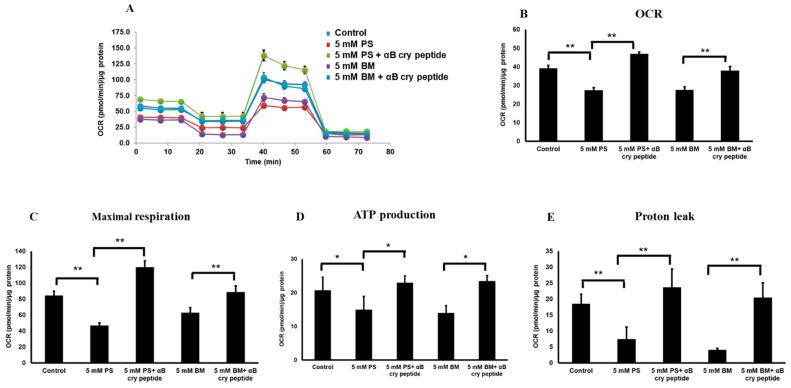
Inhibition of mGSH transporters OGC and DIC by PS and BM regulate mitochondrial bioenergetics in RPE cells. (**A**) Real-time analyses of respiratory parameters in RPE cells under OGC and DIC inhibition with or without αB cry peptide treatment. (**B**–**E**) Analysis of basal respiration, spare respiratory capacity, proton leak and ATP production, maximal respiration, and ATP production following OGC and DIC blockade in the presence or absence of αB cry peptide. A significant decrease in basal respiration was observed in DIC, and OGC inhibited cells along with a simultaneous decrease in ATP production. αB cry peptide co-treatment significantly increased mitochondrial respiration and ATP production. Data are mean ± SEM. *n* = 10; **p* < 0.1; ***p* < 0.01.

**Figure 6 antioxidants-09-00411-f006:**
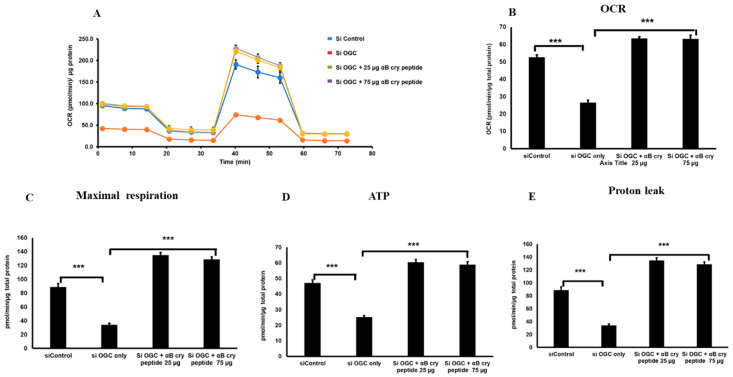
OGC knockdown in RPE caused a decline in mitochondrial oxygen consumption and ATP production. Mitochondrial bioenergetics analyses of RPE cells were made with or without OGC silencing and effect of αB cry peptide treatment was determined (**A**–**E**). SiRNA mediated OGC silencing caused a significant decrease in mitochondrial respiration, proton leak, and ATP production. Treatment of OGC silenced cells with αB cry peptide for 24 h enhanced mitochondrial respiration and associated parameters when compared to silenced controls. Data are mean ± SEM. *n* = 10; ****p* < 0.01.

**Figure 7 antioxidants-09-00411-f007:**
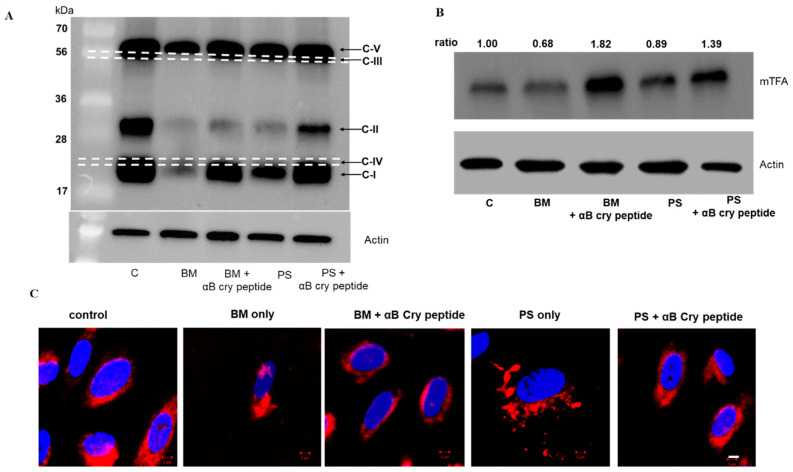
Effect of OGC and DIC inhibition on ETC proteins and mitochondrial morphology of RPE. (**A**) RPE cells were treated with and without αB cry peptide (75 µg/mL) for 24 h and total cellular protein was analyzed by Western blot under non-reducing conditions for ETC proteins using a commercial ETC complex cocktail antibody. mGSH transport inhibition resulted in decreased expression of complex proteins I and II and αB cry peptide treatment upregulated these protein subunits. Please note that the bands for complexes I and IV appeared overlapping each other making the interpretation of changes in complex I expression difficult. Complexes I and IV and complexes III and V are separated by dotted white lines in the figures for easy visualization. (**B**) Protein expression of mtTFA was measured by Western blot analysis. PS or BM treatment reduced mTFA expression, whereas co-treatment with αB cry peptide upregulated mTFA expression (**C**). RPE cells were grown on 4-well chamber slides and without αB cry peptide (75 µg/mL) for 24 h. At the end of the treatment, cells were incubated with MitoTracker Red (100 nM) for 10 min. The samples were washed, fixed in 4% PFA and viewed under LSM 770 confocal microscope. Mitotracker (Red) and DAPI (Blue) staining of RPE cells with and without treatment. In RPE cells treated with PS (5 mM) or BM (5 mM), mitochondria appeared fragmented while in the αB cry peptide group mitochondria are perinuclear, similar to that seen in control cells. These data show that inhibition of either OGC or DIC results in an altered mitochondrial arrangement. Scale bar = 5 µm.

**Figure 8 antioxidants-09-00411-f008:**
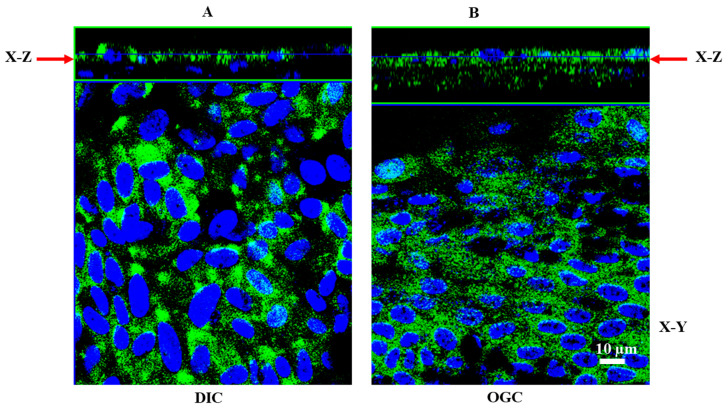
Localization of DIC and OGC in polarized RPE monolayers. RPE monolayers grown on Transwell filters were stained for OGC and DIC. The specimen was viewed on an LSM 770 laser-scanning microscope. (**A**) immunofluorescence staining of DIC in RPE monolayers showing predominant apical staining (X–Z plane). However, OGC has a wide distribution, expressed both in the apical as well as basolateral domains (**B**).

**Figure 9 antioxidants-09-00411-f009:**
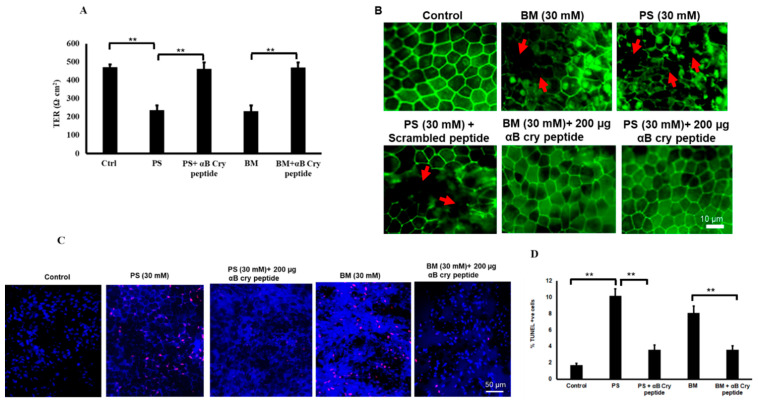
Inhibition of mGSH carriers results in loss of TER and junctional proteins in RPE which is rescued by αB cry peptide. (**A**) TER was significantly decreased when both OGC and DIC were inhibited. Coincubation with αB cry peptide (200 µg/mL) restored TER almost close to that of the TER of control cells. (**B**) Inhibiting OGC or DIC significantly decreased TER of polarized RPE monolayers. LSM confocal microscopy images showing breaks (arrows) in ZO-1 staining with 30 mM PS or BM. Co-treatment with αB cry peptide (200 µg/mL) prevents ZO1 damage (** *p* < 0.01 represents DIC or OGC inhibited vs. αB cry peptide treated monolayers). (**C**). Polarized RPE cells were treated with PS or BM (30 mM) for 24 h and cell death determined by TUNEL assay. (**D**) quantification of TUNEL positive cells. Inhibition of mGSH transporters OGC and DIC significantly increased percentage of apoptotic cells. Cotreatment with 200 µg/mL αB cry peptide significantly prevented cell death. ***p* < 0.001.

**Table 1 antioxidants-09-00411-t001:** List of antibodies used.

Antibody	Source	Application (s) & Dilution	Vendor & Product No.
OGC (Anti-SLC25A11)	Rabbit	Western blotting (1:1000) Immunofluorescence (1:100)	Abcam (ab155196)
DIC (anti-SLC25A10)	Rabbit	Immunofluorescence (1:100)	GeneTex (GTX79240)
ZO-1	Rabbit	Immunofluorescence (1:100)	ThermoFischer Scientific (61-7300)
mtTFA	Mouse	Western blotting (1:1000)	Santa Cruz Biotech (sc-376672)
Total OXPHOS Antibody Cocktail	Mouse	Western Blotting (1:1000)	Abcam (ab110411)
Actin	Mouse	Western blotting (1:2000)	Santa Cruz Biotech (sc-8432)

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
