# Peer review of "Transporter-Mediated Mitochondrial GSH Depletion Leading to Mitochondrial Dysfunction and Rescue with αB Crystallin Peptide in RPE Cells"

_antioxidants, 2020, doi:10.3390/antiox9050411_

Round 1

Reviewer 1 Report

Mitochondrial redox homeostasis is critical for cellular survival. The small tripeptide glutathione takes an important role in mitochondrial redox control. To replenish glutathione, it has to become imported from the cytosol. The transporters DIC and OGC have been suggested to facilitate glutathione transport across the IMM mainly based on pharmacological and reconstitution studies. The role of these transporters for mitochondrial glutathione homeostasis has been studied in different cells and tissues. The present study addresses their role in RPE cells, and essentially confirms their important role for mitochondrial function and cellular survival. Additionally, the authors employ the alphaB cry peptide to complement dysfunctions occurring through DIC and OGC impairment.

While the study is technically sound and well performed, it is in my opinion mainly descriptive and in parts confirmatory (i.e. presence of DIC and OGC in RPE cells; inhibition of both transporters leads to increased cell death, etc). The major concern is that like in so many other studies on DIC and OGC, there are no genetic experiments, no experiments on isolated mitochondria, no precise measurement of selectively mitochondrial glutathione. Instead all experiments are interpreted in a way as if both proteins would only and selectively transport glutathione, and the possibility that other metabolites might be affected and lead to the observed outcome is not considered. Likewise, the specificity of both, inhibitors and the alphaB cry peptide is not absolute and at least part of the results might also stem from e.g. inhibition of mitochondrial import or activation of mitochondrial proteases.

Further points:

1/ OGC and DIC also transport other metabolites. The discussion is very much tilted towards a discussion on their role in glutathione transport and neglects any other function that these proteins might have. Likewise – have the inhibitors ever been characterized with respect to a role in directly inhibiting the respiratory chain or protein import (i.e. on isolated mitochondria)?

2/ Why only silence OGC and not also DIC?

3/ What is the mechanism of action of the alphaB cry peptide in the context of mitochondrial metabolite transport? – Competition for inhibitor binding site? Titration of inhibitor vs peptide. Import of radioactive glutathione into isolated mitochondria

4/ Are the observed effects synergistic with depletion of TRR2 or genetic modulation of mitochondrial glutathione reductase levels (cytosolic and mitochondrial Glr are synthesized from the same gene, one with and one without MTS à CRISPR knockout and differential complementation; or CRISPR point mutation of first start codon)

5/ Figure 7A: what is with complexes III and IV?

6/ Introduction: “It has been confirmed that out of the eight protein carriers that are known to reside in the inner mitochondrial membrane, the DIC and OGC act as main mitochondrial GSH transporters [16,32,33].” – Only 8 protein carriers? In the IMM, there are any more metabolite/ion transport proteins

7/ Small typos throughout the text. Please carefully re-read the text.

Reviewer 2 Report

The authors proved that the addition of the αB cry peptide into hRPE cells reduces the negative effects of inhibition of two inner mitochondrial membrane anion transporters responsible for GSH transport into mitochondria. αB cry peptide restored mGSH, attenuated apoptosis, upregulated ETC proteins, improving mitochondrial bioenergetics. Also they showed that mGSH depletion lead to compromised barrier function which was partially restored by αB cry peptide. In my opinion, this is a strong article and the work was done at a high methodological level.

Few comments.

Line 2-4. Please add “in RPE cells”

Line 18-19 Please add “mitochondrial membrane anion transporters”

Figure 8-9 Please check scale bars. It seems to me that the scale  bar on F9B - 5 mkm equal to cell diameter contradicts with bar on F8 10 mkm equal to nucleus diameter

Round 2

Reviewer 1 Report

The authors have addressed my comments in writing.